# Relationship between Nutrition-Related Problems and Falls in Hemodialysis Patients: A Narrative Review

**DOI:** 10.3390/nu14153225

**Published:** 2022-08-06

**Authors:** Nobuyuki Shirai, Tatsuro Inoue, Masato Ogawa, Masatsugu Okamura, Shinichiro Morishita, Yamamoto Suguru, Atsuhiro Tsubaki

**Affiliations:** 1Department of Rehabilitation, Niigata Rinko Hospital, Niigata 950-8725, Japan; 2Department of Physical Therapy, Niigata University of Health and Welfare, Niigata 950-3198, Japan; 3Division of Rehabilitation Medicine, Kobe University Hospital, Kobe 650-0017, Japan; 4Berlin Institute of Health Center for Regenerative Therapies (BCRT), Charité–Universitätsmedizin Berlin, 13353 Berlin, Germany; 5Department of Physical Therapy, School of Health Science, Fukushima Medical University, Fukushima 960-1295, Japan; 6Division of Clinical Nephrology and Rheumatology, Niigata University Graduate School of Medical and Dental Sciences, Niigata 951-8520, Japan

**Keywords:** falls, hemodialysis, frailty, sarcopenia, undernutrition

## Abstract

Falls are a social problem that increase healthcare costs. Hemodialysis (HD) patients need to avoid falling because fractures increase their risk of death. Nutritional problems such as frailty, sarcopenia, undernutrition, protein-energy wasting (PEW), and cachexia may increase the risk of falls and fractures in patients with HD. This review aimed to summarize the impact of frailty, sarcopenia, undernutrition, PEW, and cachexia on falls in HD patients. The reported global incidence of falls in HD patients is 0.85–1.60 falls per patient per year. HD patients fall frequently, but few reports have investigated the relationship between nutrition-related problems and falls. Several studies reported that frailty and undernutrition increase the risk of falls in HD patients. Nutritional therapy may help to prevent falls in HD patients. HD patients’ falls are caused by nutritional problems such as iatrogenic and non-iatrogenic factors. Falls increase a person’s fear of falling, reducing physical activity, which then causes muscle weakness and further decreased physical activity; this cycle can cause multiple falls. Further research is necessary to clarify the relationships between falls and sarcopenia, cachexia, and PEW. Routine clinical assessments of nutrition-related problems are crucial to prevent falls in HD patients.

## 1. Introduction

Falls are a social problem that increase medical expenses. In 2015, the estimated medical costs due to falls in the United States were approximately USD 50 billion [1]. According to the World Health Organization, approximately 28–35% of the elderly community fall every year [2]. Falls can cause fractures and head injuries [3], poor quality of life [4], early admission to long-term care facilities [5], and increase the mortality rate [6]. In addition, falls cause a fear of falling, thereby decreasing physical activity [7].

Hemodialysis (HD) patients frequently fall [8] which leads to fractures and increased mortality. The global number of people receiving renal replacement therapy was reported to be 262 million in 2010, and is expected to increase to 543.9 million by 2030 [9]. HD patients are at a greater risk than healthy individuals of developing femoral neck fractures because of decreased bone mineral density due to progressive chronic kidney disease (CKD) [10,11]. Mortality in HD patients with fractures are higher than in those without fractures [12]. Therefore, fall prevention is necessary for HD patients in order to decrease the economic and social issues.

Nutritional problems may be a risk factor for falls in HD patients. HD patients develop symptoms of uremia, decreased dietary intake and protein synthesis, increased catabolism, metabolic acidosis, chronic inflammation, and insulin resistance [13]. HD treatment causes a loss of nutrients and decreased physical activity. As a result, frailty, sarcopenia, and undernutrition are accelerated due to a decrease in energy storage sources such as protein, muscle, and fat [13]. Nutrition-related disorders are associated with falls in the elderly community [14,15,16]. However, the relationship between nutritional problems and falls in HD patients remains unclear.

The purposes of this narrative review were to summarize (1) the causes of falls in HD patients, and (2) the impact of frailty, sarcopenia, undernutrition, protein-energy wasting (PEW), and cachexia on falls in HD patients. Clarifying the effects of nutritional problems on falls may help to prevent falls in HD patients.

## 2. Definition of Falls

According to The Prevention of Falls Network Europe Consensus, a fall is defined as “an unexpected event in which the participants come to rest on the ground, floor, or lower level” [17]. Falls should be recorded using prospective daily recording and a notification system with a minimum of monthly reporting. Telephone or face-to-face interviews should be used to rectify missing data and to ascertain further details of falls and injuries [17]. Fall data should be summarized as the number of falls, number of fallers/non-fallers/frequent fallers, fall rate per person per year, and first fall [17].

## 3. Incidence, Trigger, and Timing of Falls in HD Patients

### 3.1. Frequency of Falls in HD Patients

HD patients had far more falls than the general population. Fifteen studies on falls in HD patients were extracted (Table 1). Approximately 30–60% of HD patients experience falls [18,19,20,21,22,23,24,25,26], and of these, 30–57% of patients had multiple falls [18,21,24,25,26]. The rate of severe falls requiring medical attention was 10.7–19.0% [18,26,27] and fractures occurred in 1.0–4.0% [18,26,28]. HD patients (mean age of 70 years) accounted for 45.0–55.5% of falls annually, compared with approximately 30% in the elderly community [29]. The reported global incidence of falls in HD patients was 0.85–1.60 falls per patient–year. In addition, falls reportedly increased with age to 1.76 falls per patient–year for those ≥65 years old compared to 0.13 falls per patient–year for those <65 years old [30]. There is no significant difference between peritoneal dialysis (PD) patients and HD patients with regard to fall frequency (OR 1.63; 95% CI 0.88–3.04; *p* = 0.12) after adjusting for covariates [23]. Van Loon et al. reported a median number of falls was two (Interquartile range IQR) [1,2,3,4,5], which was comparable between HD and PD [31]. 

### 3.2. Trigger for Falls in HD Patients

Falls often occur when walking at home in patients with HD. Cook et al. reported that walking was the most common activity that led to falls (69% indoors, 31% outdoors) [18]. They also reported frequent falls in the process of moving from the sitting position to the standing position (31%) and moving from a recumbent position to the standing position (12%) [18]. Zanotto T et al. reported that the causes of falls were gait and balance issues (65.4%), environmental hazards (46.2%), and dizziness or syncope-like events (42.3%) [24]. In addition to walking (31%), falls occurred when getting up (21%), turning around (15%), using stairs (6%), and others (26%) [26]. The locations of falls are at home (72–82%), outdoors (19%), public sites (7–9%), and others (9%) [26,28].

### 3.3. Timing of Falls in HD Patients

Falls tend to occur more often after than before dialysis sessions, but no significant difference has been reported when comparing the days on which patients underwent dialysis and the days when they did not. Desmet et al. reported that the frequency of falls was higher within 22 h after dialysis therapy than within 22 h before dialysis therapy, but the difference was not significant (*p* = 0.058) [28]. Furthermore, Cook et al. reported that falls occurred at similar frequencies on dialysis and non-dialysis days (*p* = 0.05) [18]. The average number of falls per person was 1.45 (95% CI 0.89–2.01) on dialysis days and 1.35 (95% CI 0.82–1.89) on non-dialysis days after adjusting for the weekly dialysis frequency [18].

### 3.4. Falls and Clinical Outcomes in HD Patients

Falls in HD patients cause poor clinical outcomes. Falls resulted in hospitalization for 16% and death for 4% of instances [18]. Abdel-Rahman et al. reported that the faller had a 2.13-times increased risk of death, 3.5-times increased risk of admission to a nursing home, and a nearly two-fold increase in the number and duration of hospitalizations [21].

## 4. Nutritional Problems and Falls in HD Patients

### 4.1. Frailty and Falls in HD Patients

Frailty is a condition that increases vulnerability due to increased dependency when exposed to a stressor [34]. The frailty phenotype is the most commonly used evaluation tool for frailty, and consists of five symptoms: weakness, slow walking speed, low physical activity, exhaustion, and unintentional weight loss [35]. Frailty is diagnosed if three or more of the symptoms are present [35]. The prevalence of frailty in HD patients ranges from 29.6–81.5% [36]. The prevalence of frailty with end-stage renal disease (ESRD) is 46.0% (95% CI, 34.2–58.3%) [36]. The risk factors of frailty are age (SMD, 0.43 years; 95% CI, 0.24–0.61), female sex (OR, 1.89; 95% CI, 1.33–2.67), and diabetes (OR, 2.42; 95% CI 1.68–3.49) [36].

The prevalence of frailty in HD patients ranges from 29.6–81.5% [36]. The prevalence of frailty with end-stage renal disease (ESRD) is 46.0% (95% CI, 34.2–58.3%) [36]. The risk factors of frailty are age (SMD, 0.43 years; 95% CI, 0.24–0.61), female sex (OR, 1.89; 95% CI, 1.33–2.67), and diabetes (OR, 2.42; 95% CI 1.68–3.49) [36].

Frailty increases the risk of falls in HD patients (Table 2). Frail participants experienced increased falls 3.09-fold (95% CI, 1.38–6.90) compared to non-frail participants, after adjusting for covariates. [32]. Chu et al. reported that the prevalence of frailty in kidney transplant candidates was 1.36 times (95% CI, 1.12–1.64) higher in the single fall group and 1.90 times (95% CI, 1.58–2.29) higher in the recurrent falls group compared to the non-fall group. In addition, the prevalence of frailty in kidney transplant recipients was 1.67 times (95% CI, 1.02–2.74) higher in the single fall group and 2.04 times (95% CI, 1.20–3.45) higher in the recurrent fall group compared to the non-fall group [27]. Delgado et al. reported that self-reported frailty was associated with a higher risk of falls or fractures requiring medical attention compared to non-frail participants (HR, 1.60; 95% CI, 1.16–2.20) [37]. Thus, frailty causes falls, and interventions for frailty may lead to the prevention of falls.

### 4.2. Sarcopenia and Falls in HD Patients

HD patients are more likely to have sarcopenia, which is characterized by the age-related loss of skeletal muscle plus low muscle strength and/or physical performance [38]. The definition of sarcopenia is recommended by the Asian Working Group for Sarcopenia (AWGS) [39] for Asians and the European Working Group on Sarcopenia in Older People (EWGSOP) [40] in Europe and the United States. Both include the clinical features loss of skeletal muscle mass, low muscle strength, and low physical performance [39,40]. The reported prevalence of sarcopenia ranges from 4–42% in patients with HD, depending on the diagnostic criteria [41]. Bataille et al. reported that 31.5% of HD patients had sarcopenia, 33.3% had low muscle mass, and 88.3% had low muscle strength [42]. Protein catabolism due to the dialysis procedure, and a low energy and protein intake leads to sarcopenia in patients with HD [43,44]. Inactivity and fatigue during dialysis treatment [45] cause a loss of muscle mass, muscle weakness, and decreased physical function.

No studies have reported on the association between sarcopenia and falls in HD patients to date. However, lower skeletal muscle mass, assessed by the modified creatinine index (MCI), leads to fractures in HD patients. MCI is calculated by the following formula: MCI (mg/kg/d) = 16.21 + 1.12 × [one if male; zero if female] − 0.06 × age (years) − 0.08 × single-pool Kt/V for urea + 0.009 × serum creatinine before dialysis (mmol/L) [46]. Considering the MCI quartiles Q1 (HR, 7.81; 95% CI, 2.63–23.26) and Q2 (HR, 5.48; 95% CI, 2.08–14.40) compared to Q4, the fracture rates in men were significantly higher [46]. Comparing the MCI quartile Q1 (HR, 4.44; 95% CI, 1.50–13.11) to Q4, the fracture rates in women were significantly higher [46].

The early detection of sarcopenia is important because HD patients have a high prevalence of sarcopenia. However, it is difficult to diagnose sarcopenia by the measurement of muscle mass due to the need for computed tomography (CT) and dual-energy X-ray absorptiometry (DEXA), which require expensive equipment, and also due to the radiation to which the patients would be exposed. The MCI can be calculated from age, gender, Kt/V, and blood creatinine levels [46]. The MCI has also been shown to significantly correlate with skeletal muscle mass measured by bioelectrical impedance analysis (BIA) [46]. Therefore, further research into falls in HD patients related to sarcopenia is necessary.

### 4.3. Undernutrition and Falls in HD Patients

Undernutrition is common in patients with HD and is caused by the progression of CKD [47], a low protein diet [48], and prolonged dialysis treatment after HD introduction [49]. Undernutrition is classified as iatrogenic and non-iatrogenic [50]. Iatrogenic factors are the result of dialysis treatment. The loss of nutrients from dialysis treatment is 3–8 g of amino acids and 3–9 g of protein per day [50]. Reusing the dialyzer can significantly reduce albumin in the dialysate and significantly change the permeability of the dialyzer [51]. Loss of amino acids differ depending on the dialysis method, such as the type of dialyzer membrane. On the other hand, non-iatrogenic factors occur spontaneously from factors associated with the progression of CKD, such as a loss of appetite and decreased physical function [50]. Many nutritional indicators and screening tools are used to assess the nutritional status in HD patients, as exemplified by the body mass index (BMI) [49], brachialis muscle circumference [52], albumin [49], diet energy intake [52], fat tissue index (FTI) [53], lean tissue index (LTI) [53], Subjective Global Assessment (SGA) [54], and Geriatric Nutritional Risk Index (GNRI) [33,55].

Undernutrition is a critical predictor of falls in HD patients (Table 2). Kono et al. reported that the GNRI was lower in fallers than in non-fallers (HR, 1.04; 95% CI, 1.01–1.08) [33]. Undernutrition defined by weight loss was significantly higher in patients who experienced severe falls than in non-fallers (OR, 8.4; 95% CI, 1.7–42.4) [56]. Decreased serum albumin was independently associated with an increased risk of fractures (HR, 0.87; 95% CI, 0.77–0.91) in HD patients [57]. A low GNRI (<92) was also significantly associated with an increased risk of fractures, even after adjusting for confounding factors (male HR, 2.93; 95% CI 1.54–5.59; female HR, 2.05; 95% CI 1.20–3.51) [58]. Therefore, the risk of falls and fractures in HD patients may be reduced by approach or interventions for undernutrition.

PEW is a state of nutritional and metabolic disorders in patients with CKD and end-stage renal disease characterized by the simultaneous loss of systemic body protein and energy storage [59]. PEW is defined as (1) abnormal serum chemistry such as hypoalbuminemia; (2) a loss of body mass (BMI, weight loss, body fat percentage); (3) a loss of muscle mass (muscle wasting area around the brachialis muscle, low creatinine appearance); (4) a decrease in the creatinine production rate, and a decrease in dietary intake (an unintentional decrease in protein intake, unintentional decrease in energy intake). PEW is diagnosed when there is at least one item in ≥three of the four categories [59]. The prevalence of PEW is 28–54% in HD patients [60].

The association between PEW and falls in HD patients is unclear. HD patients with increased risk of frailty, inadequate nutrition, and impaired bone mineral content are likely to be associated with fractures [61]. Therefore, PEW may be associated with falls and fractures.

### 4.4. Cachexia and Falls in HD Patients

Cachexia is a condition that combines skeletal muscle mass loss and weight loss due to complex metabolic disorders. It is defined as a complex metabolic syndrome associated with underlying illness, and is characterized by a loss of muscle mass with or without a loss of fat mass [62]. The prominent clinical feature of cachexia in adults is weight loss (corrected for fluid retention), and in children is growth failure (excluding endocrine disorders). Anorexia, inflammation, insulin resistance, and increased muscle protein breakdown are also frequently associated with cachexia [62]. The diagnostic criterion is weight loss of at least 5% in 12 months or less (or BMI < 20 kg/m^2^) in the presence of underlying disease, and is confirmed if three or more of the following five criteria are met: (1) decreased muscle strength; (2) fatigue; (3) anorexia (total calorie intake less than 20 kcal/kg body weight/day, less than 70% of normal food intake); (4) low fat-free mass index (lean body mass); and (5) abnormal biochemistry (Alb < 3.2 g/dL, Hb < 12.0 g/dL, C-reactive protein (CRP) > 0.5 mg/dL) [62].

The prevalence of cachexia in HD patients may be higher than that in patients with other diseases. The prevalence of cachexia in HD patients is 16% [63] and 30–60% in CKD patients [64]. Cachexia is referred to as a severe form of PEW, often associated with profound physiological, metabolic, psychological, and immunological disorders. In addition, it is a severe form of metabolic depletion, whereas PEW can also be used to refer to mild degrees of depleted protein and energy mass [59]. Further studies are needed to investigate the association between cachexia and falls in HD patients.

## 5. Other Risk Factors for Falls in HD Patients

Falls in HD patients are influenced by many factors other than nutritional disorders. The accumulation of uremic substances leads to abnormalities in muscle cells and is a cause of muscle atrophy [64,65]. Diabetes is also one of the risk factors for CKD and is the most common primary disease leading to dialysis [66]. Diabetic neuropathy and diabetic retinopathy cause decreased proprioceptive sensibility, orthostatic hypotension due to autonomic neuropathy, low leg muscle strength, and impaired visual acuity [67,68]. Furthermore, diabetic nephropathy and anemia are related to a vitamin B12 deficiency or metformin treatment [67]. In addition, a vitamin D deficiency is more common in CKD and HD patients [69]. Decreased vitamin D has been shown to be associated with falls and decreased physical function in older adults [70]. Dialysis-related factors can also increase the risk of falls. HD patients may experience a deterioration of balance function due to decreased blood pressure and muscle blood flow due to water removal [71]. HD patients lose muscle mass due to the loss of nutrients involved in muscle metabolism [72,73] and the loss of physical activity during HD [74].

Many factors have been reported to be associated with falls in HD patients (Table 1). Age [21,33], gender [18,21,23], fall history [18,23,31], high risk of falls [56], antidepressant use [28], polypharmacy [28], and low quality of life [27] were independently associated with falls. The lower limb muscle strength and physical function were also risk factors for falls. A reduced grip strength [33], low short physical performance battery (SPPB) score [27,33], gait disturbance [28], and balance dysfunction in a static standing position with eyes closed [25] were independently associated with falls. CKD and dialysis-related factors were also risk factors for falls. Diabetes [28,31], the number of complications [15,23], elevated parathyroid hormone [22], high CRP [33], low pre-HD mean systolic blood pressure [18,22], dialysis-related hypotension [33], orthostatic hypotension [26], and the deterioration of the arterial baroreflex function [26] were independently associated with falls. CKD-related factors due to renal dysfunction and dialysis therapy may increase the risk of falls. The progression of frailty syndrome, decreased balance function, and decreased blood pressure directly or indirectly affect HD patients.

## 6. Fall Prevention Strategies for Nutrition-Related Problems in HD Patients

Frailty is a risk factor for falls and nutritional therapy may prevent falls in HD patients. Figure 1 shows the flow to falls due to nutritional problems in HD patients. HD patients experience nutritional problems such as frailty, undernutrition, sarcopenia, PEW, and cachexia due to iatrogenic and non-iatrogenic factors [50]. Falls increase the fear of falling, thus reducing physical activity, causing muscle weakness and further decreased physical activity [31,74]; this cycle causes multiple falls.

Nutritional management is important for HD patients to maintain muscle mass. Individual nutritional counseling, the optimization of dialysis regimens, prevention or amelioration of muscle wasting, and comorbidities (metabolic acidosis, diabetes, infections, depressive heart failure, depression, and others) are needed for HD patients. If these practices cannot maintain protein and energy storage, oral or parenteral nutrition should be prescribed along with appetite stimulants and muscle-building agents [75]. The oral protein-based dietary supplement group of HD patients were reported to have improved serum prealbumin levels and central arm circumference compared to the placebo or supplement medicine group [76].

A combination of physical therapy and nutritional therapy can prevent falls. As a result of long-term exercise and amino acid supplementation in the elderly in the community, the non-intervention group had an increased fall rate, but the fall rate in the intervention group did not change significantly [77]. Physical therapy can prevent falls by improving the physical function of the lower limbs [78]. Adequate physical therapy based on an individual‘s physical function and activity levels is important for managing physical weakness in HD patients [79]. In recent years, it has become clear that exercise therapy during dialysis is effective in improving physical function [80]. However, HD patients are known to have low exercise compliance [81]. Exercise therapy may be beneficial to all of these patients [82]. It may be a useful idea to start with standing exercises in the dialysis room or waiting room [83]. Lower limb muscle strength is associated with physical activity and is also important for HD patients [84]. The recommended amount of physical activity for HD patients is 4000 steps/day or more on non-dialysis days [85]. The elderly in the community were included in multiple exercise categories for preventing primary falls [86]. In particular, there has been robust evidence that the combination of balance and functional exercise was beneficial for the prevention of falling [86]. Therefore, HD patients can prevent the risk of falls by incorporating walking training and balance training.

## 7. Conclusions

There have been few reports of nutrition-related problems and falls in HD patients. Frailty and undernutrition in HD patients are strong risk factors for an increase in the number of falls. Further research is necessary to clarify the relationships between falls and sarcopenia, cachexia, and PEW. Routine clinical assessments of nutrition-related problems are crucial to prevent falls in HD patients.

## Figures and Tables

**Figure 1 nutrients-14-03225-f001:**
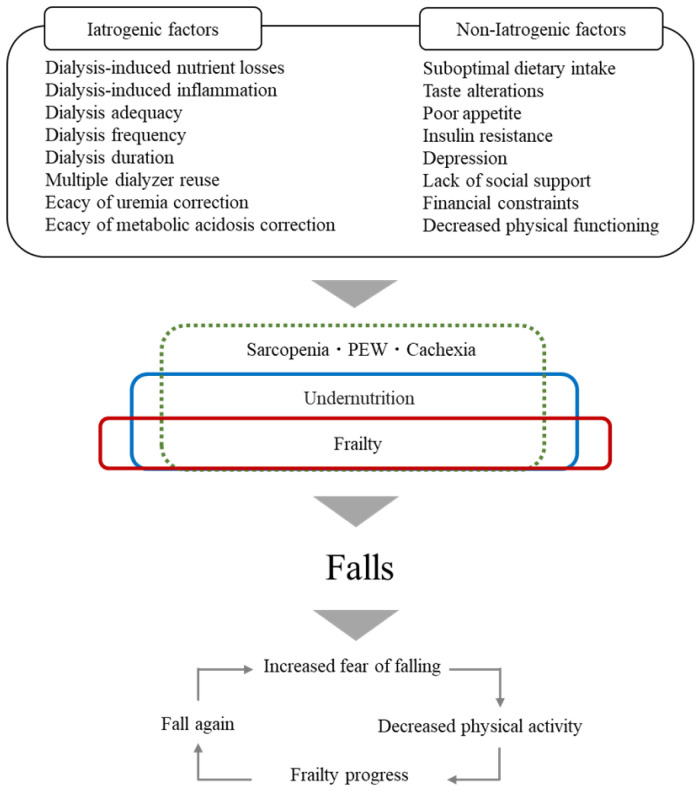
Flow to falls due to nutritional problems in HD patients.

**Table 1 nutrients-14-03225-t001:** Frequency and factors of falls in HD patients.

Author, Year, Country	Design, Setting (Study Period)	Age (Years) Male/Female	Sample Size, Falls (First Fall)	Fall Activities, Location	Fall Timing	Fall Risk Factors
Roberts RG et al., 2003 [19] UK	Cross-sectional study One hospital HD unit	Mean 78.2 ± 5.3 Male 48.9% Female 51.1%	*n* = 47 13 (27.7%)	Not listed	Not listed	Postural hypotension ※No multivariate analysis
Desmet C et al., 2005 [28] Belgium	Prospective cohort study (8 weeks) Seven HD units	Median 70.9 (25.3–92.8) Male 56.4% Female 43.6%	*n* = 308 56 (12.7%)	Home (82%), Public sites (7%), Other (9%), unknown (2%)	Five falls were recorded during the first 24 h after dialysis. Among the remaining 33 falls, 24 falls were observed within 22 h after HD therapy versus 9 falls within 22 h before HD therapy (*p* = 0.058).	Older age, diabetes, walking test failed, intake of an antidepressant, and high number of oral prescribed drugs
Cook WL et al., 2006 [18] Canada	Prospective cohort study (12 months) One outpatient HD unit	Mean 74.7 Male 57% Female 43%	*n* = 169 305 (45.0%)	Walking (69% indoors, 31% outdoors), standing from the seated position (31%), trying to rise from a lying position (12%).	Falls occurred with similar frequency on dialysis and non-dialysis days (*p* = 0.05). On dialysis days, falls were more common after dialysis (73%) than before (27%).	Male gender, history of falls, low mean pre-dialysis systolic blood pressure, higher number of comorbidities
Roberts R et al., 2007 [30] UK	Prospective cohort study (6 months) One hospital HD unit	Median 58 (52–83) Male 65.4% Female 34.6%	*n* = 78 14 ・aged > 65: (38%) ・younger: (4%)	Not listed	Not listed	Age ※No multivariate analysis
Li M et al., 2008 [20] Canada	Prospective cohort study (12 months) One hospital HD unit	Mean 74.7 ± 6.1 Male 57.0% Female 43.0%	*n* = 162 305 (46.9%)	Not listed	Not listed	Older, higher number of comorbidities, diabetes, initiated renal replacement therapy more recently ※No multivariate analysis
Abdel-Rahman EM et al., 2010 [21] USA	Prospective cohort study (12 months) Two outpatient HD units	Mean 62.4 ± 16.1 Male 61.8% Female 38.2%	*n* = 76 20 (26.3%)	Not listed	Not listed	Age ≥ 65 years, Female
McAdams-DeMarco MA et al., 2013 [32] USA	Prospective cohort study (6.7 months) One outpatient dialysis unit	Mean 65 ± 12.6 Male 53.7% Female 46.3%	*n* = 95 70 (28.3%)	Not listed	Not listed	Frailty
Polinder-Bos HA et al., 2014 [22] Netherlands	Prospective cohort study (12 months) Two hospital HD units	Median 79.3 (70–89) Male 52% Female 48%	*n* = 49 40 (55.5%)	Not listed	Not listed	Lower systolic blood pressure before dialysis, higher PTH
Farragher JF et al., 2016 [23] Canada	Prospective cohort study (12 months) One hospital HD unit	・HD Mean 74.7 ± 6.1 Male 57.0% Female 43.0% ・PD Mean 73.2 ± 9.0 Male 55.0% Female 45.0%	・HD *n* = 162 305 (46.9%) ・PD *n* = 74 87 (54%)	Not listed	Not listed	Male, number of comorbidities, ≥1 reported fall in previous year
Zanotto T. et al., 2018 [24] UK	Cross-sectional study Two hospital HD units	Mean 61.1 ± 14 Male 53.9% Female 46.1%	*n* = 72 80 (36.1%)	Gait and balance issues (65.4%), environmental hazards (46.2%), and dizziness or syncope-like events (42.3%).	Not listed	None of the variables were significantly associated with falling.
Kono K. et al., 2018 [33] Japan	Prospective cohort study (2 years) Two outpatient dialysis units	Mean 69.4 ± 11.6 Male 60% Female 40%	*n* = 223 91(41%)	Not listed	Not listed	Age of 80 years and older, high CRP level, decreasing GNRI, SPPB 8 points or less, decreasing grip, presence of intradialytic hypotension, high scores in the inquiry regarding falling
van Loon IN et al., 2019 [31] England and Northern Ireland	Prospective cohort study (24 months) Twenty-two outpatient HD units	Mean 75.0 ± 7.0 Male 60.0% Female 40.0%	*n* = 203 54 (47%)	Not listed	Not listed	Diabetes mellitus, previous falls
Zanotto T. et al., 2020 [25] UK	Prospective cohort study (12 months) Three outpatient HD units	Mean 61.8 ± 13.4 Male 54.4% Female 45.6%	*n* = 68 25 (36.8%)	Not listed	Not listed	Higher center of pressure range in medial–lateral direction during eyes closed
Zanotto T et al., 2020 [26] UK	Prospective cohort study (12 months) Three hospital HD units	Mean 61.7 ± 13.3 Male 55.1% Female 44.9%	*n* = 69 80 (37.7%)	Walking (31%), getting up (21%), turning around (15%), using stairs (6%), other (26%). Home (72%), outdoors (19%), public site (9%).	Not listed	Worse baroreflex function, orthostatic decrements of blood pressure to 60° head-up tilt test
Chu NM et al., 2020 [27] USA	Prospective cohort study (108 months) Two hospital HD units	・Kidney transplantation donors Mean 54.0 ± 14.0 Male 61.9% Female 38.1% ・Kidney transplantation recipients Mean 54.3 ± 14.0 Male 62.1% Female 37.9%	・Kidney transplantation donors *n* = 3666 598 (16.3%) ・Kidney transplantation recipients *n* = 770 96 (12.5%)	Not listed	Not listed	・Kidney transplantation candidates: frailty, lower extremity impairment (SPPB score ≤ 10), poor HRQOL ・Kidney transplantation recipients: frailty, lower extremity impairment (SPPB score ≤ 10)

HD, hemodialysis; PTH, parathyroid hormone; PD, peritoneal dialysis; CRP, C-reactive protein; GNRI, geriatric nutritional risk index; SPPB, Short Physical Performance Battery; HRQOL, health-related quality of life.

**Table 2 nutrients-14-03225-t002:** Nutritional problems and falls in HD patients.

Author, Year, Country	Design, Setting (Study Period)	Age (Years) Male/Female (%)	Sample Size, Falls (First Fall)	Evaluation	Main Results
McAdams-DeMarco MA. et al., 2013 [32] USA	Prospective cohort study (6.7 months) One outpatient dialysis unit	Mean 65 ± 12.6 Male 53.7% Female 46.3%	*n* = 95 70 (28.3%)	Fried frailty phenotype	After adjusting for comorbidities, disability, number of medications, education, and marital status, frailty predicted a 3.09-fold (95% CI: 1.38–6.90, *p* = 0.006) higher number of falls.
Chu NM et al., 2020 [27] USA	Prospective cohort study (108 months) Two hospital HD units	・Kidney transplantation candidates Mean 54.0 ± 14.0 Male 61.9% Female 38.1% ・Kidney transplantation recipients Mean 54.3 ± 14.0 Male 62.1% Female 37.9%	・Kidney transplantation candidates *n* = 3666 598 (16.3%) ・Kidney transplantation recipients *n* = 770 96 (12.5%)	Fried frailty phenotype	・Kidney transplantation candidates: frailty was independently associated with single fall (PR, 1.36; 95% CI, 1.12–1.64) and recurrent falls (PR, 1.90; 95% CI: 1.58–2.29). ・Kidney transplantation recipients: frailty was independently associated with single fall (PR, 1.67; 95% CI, 1.02–2.74) and recurrent falls (PR, 2.04; 95% CI, 1.20–3.45).
Kono K. et al., 2018 [33] Japan	Prospective cohort study (2 years). Two outpatient dialysis unit	Mean 69.4 ± 11.6 Male 60% Female 40%	*n* = 223 91 (41%)	GNRI	In the univariate analysis, decreasing GNRI was independently associated with falls (HR, 1.04; 95% CI, 1.01–1.08).

HD: hemodialysis, CI: Confidence Interval, PR: prevalence ratio, GNRI: geriatric nutritional risk index, HR: hazard ratio.

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
