# Peer review of "Relationship between Nutrition-Related Problems and Falls in Hemodialysis Patients: A Narrative Review"

_nutrients, 2022, doi:10.3390/nu14153225_

Round 1

Reviewer 1 Report

This is an interesting and well-written (very comprehensive) review of the relationship between nutrition-related problems and accidental falls in hemodialysis patients. I have only few minor comments:

- line 198 - PEW is presented as an acronym, but it isn't presented what is the meaning of this abbreviation .

-line 235 - diabetes is presented as an aditional risk factor for falls. As this is a review about the nutritional risk factors, it should be mentioned that the higher risk in diabetic patients is determined by the presence of the specific complications (neuropathy with loss of proprioceptive sensibility and the presence of the ortostatic hypotension, vision loss related to the macular edema or retinopathy) or a specific alteration of the  nutritional status related to diabetes (diabetic nephropathy, anemia related with vitamine B12 deficiency  or metformin treatment). If such studies which relate diabetes with the nutritional risk factors should be mentioned.

Sincerly yours,

Author Response

添付ファイルをご覧ください。

Reviewer 2 Report

Thank you for an informative and well structured narrative review. This is an important topic and your paper does a good job of bringing this to ones attention in an easy to read and digest format. The main recommendation I have is in relation to the use of 'accidental fall' in the title and throughout. When you are defining the term (lines 62-64) you define only 'falls' not accidental falls. Are you suggesting these are different? if so please define the accidental element, if not, then I'd suggest just using 'falls' throughout.
